# Preparation and Hydration of Brownmillerite-Belite-Sulfoaluminate Cement

**DOI:** 10.3390/ma15124344

**Published:** 2022-06-20

**Authors:** Xuemei Chen, Jun Li, Zhongyuan Lu, Yunhui Niu, Jun Jiang, Yigang Xu, Wen Zhong

**Affiliations:** 1State Key Laboratory of Environment-Friendly Energy Materials, School of Material Science and Engineering, Southwest University of Science and Technology, Mianyang 621010, China; chunruxue@126.com (X.C.); niuyunhui@swust.edu.cn (Y.N.); jiangjun@swust.edu.cn (J.J.); 2Jiahua Special Cement Co., Ltd., No. Ma’anshan Jiufeng Rd., Shizhong District, Leshan 614003, China; zhuasc@163.com (Y.X.); scjhzw@163.com (W.Z.)

**Keywords:** brownmillerite-belite-sulfoaluminate cement, hydration, mechanical performance

## Abstract

Brownmillerite-belite-sulfoaluminate clinker with different contents of brownmillerite were designed and successfully prepared by using limestone (LS), aluminum tailings (AT), aluminum mine (AM), and anhydrite (AH) calcined at 1330 °C for 30 min. Then, three kinds of brownmillerite-belite-sulfoaluminate cement (BBSC) were obtained by grinding mixtures of the clinker and AH. Hydration and mechanical performances of the prepared BBSC were thus intensively studied. The increase in brownmillerite in BBSC decreased the hydration exothermic rate and delayed the renewed rapid formation of AFt at early hydration stages. However, the formation of C_2_AS·8H_2_O would be promoted, where the higher the brownmillerite content in BBSC, the earlier the C_2_AS·8H_2_O formed. The increase in brownmillerite might change the morphologies of the formed AFt, grass-shaped AFt enriched in iron would be the main hydration products in BBSC with a higher content of brownmillerite. The increase in brownmillerite content contributed to the stability improvement in flexural strength and the stable growth in the compressive strength of BBSC. The appropriate content of brownmillerite (20 wt%) can balance the whole hydration reaction process, which was conducive to the development of BBSC mechanical strength, the decrease in the hydration heat release, and the volume stability of hardened pastes.

## 1. Introduction

The consumption of fossil fuels and the decomposition of limestones contribute the most CO_2_ emissions in cement production. Developing the cement clinker with lower energy input and CaO content thus draws much more attention worldwide [1,2,3,4,5,6].

There are four main minerals, alite (C_3_S), belite (C_2_S), tricalcium aluminate (C_3_A), and tetracalcium aluminoferrite (also known as brownmillerite) in Portland cement clinker, and C_2_S has the characteristics of lower hydration heat, higher later mechanical properties, higher growth rate of mechanical properties, and excellent durability [1,2]. In addition, the formation temperature and CaO content of C_2_S are relatively lower than that of the other minerals in Portland cement clinker, that is, fewer fossil fuels and limestones are required to produce C_2_S. Therefore, high C_2_S cement has become one of the most important developing directions in green low-carbon cement [1]. However, a lower early hydration rate of C_2_S decreases the early mechanical performance of cement and concrete, thus significantly decreasing the construction rate. Recently, another low-carbon clinker mineral named anhydrous calcium sulfoaluminate (C_4_A_3_$) with fast setting, high early mechanical performance, low alkalinity, micro-expansion, and corrosion resistance [3] was proposed for introduction into a high C_2_S cement system, and the newly-formed high belite-sulfoaluminate cement system will overcome the lower early mechanical performances of high C_2_S cement [4,5,6]. The CO_2_ footprint of high belite-sulfoaluminate cement is about 30% lower than that of OPC because these materials require reduced amounts of limestone and lower operating temperatures and, in addition, they are easily ground [7]. However, high belite-sulfoaluminate cements are not yet at large-scale industrial production because they present quite low mechanical strength due to their high content of belite with slow reactivity [8]. Recently, a new class of BCSAF cement has been patented by Lafarge, in which high-temperature belite polymorphs (α-forms), stabilized by boron doping together with high early mechanical strength, were obtained [9]. Generally, aluminum materials containing lots of Fe_2_O_3_ are easy to obtain and are commonly used as the raw materials of high belite-sulfoaluminate cement production, thus, a large amount of brownmillerite phases are generated and enriched in high belite-sulfoaluminate cement clinker in this case [1,7,8,9,10,11]. High belite-sulfoaluminate cement with a relatively higher content of brownmillerite can also be named as brownmillerite-belite-sulfoaluminate cement (BBSC). Presently, the hydration of brownmillerite and the belite-sulfoaluminate containing brownmillerite has been researched [1,7,8,9,10,11,12,13,14,15,16,17,18,19]. However, the hydration and flexural/compressive strength of BBSC have rarely been clarified.

Currently, aluminum materials with relatively higher Fe_2_O_3_ are increasingly being used in high belite-sulfoaluminate cement production because of their low cost and abundant sources compared to high-quality aluminum materials. Therefore, it is important to study the role of brownmillerite in the hydration of the BBSC system. In this study, a BBSC clinker with different brownmillerite content was first designed and prepared, and then the three kinds of BBSC were obtained accordingly. The effect of brownmillerite on the properties and hydration of BBSC was also investigated.

## 2. Experimental

### 2.1. Raw Materials

Limestone (LS), aluminum tailings (AT) from bauxite mineral processing, aluminum mine (AM) from waste bauxite rock and anhydrite (AH) were all provided by Qianghua Special Cement Engineering Co. Ltd., Leshan, China. The main chemical compositions of raw materials are shown in Table 1. The sand used is ISO Standard Sand produced by Xiamen ISO Standard Sand Co. Ltd., Xiamen, China.

### 2.2. Preparation

#### 2.2.1. Design of the BBSC Clinker

The mineral compositions of BBSC clinkers were first designed and calculated. A modified Bogue equation was applied to estimate the theoretical phase compositions of the BBSC clinker [20]. The theoretical alkalinity (*C*m), a weight ratio of Al_2_O_3_/SiO_2_ (N) and Al_2_O_3_/SO_3_ (P), was calculated as shown in Equations (1)–(3). *C*m presents the content of CaO in the raw materials, which can meet the requirement of producing useful minerals in cement clinker. The other phases, C_4_AF, C_4_A_3_$, C_2_S, and CaSO_4_ (C$), were calculated as shown in Equations (4)–(7).
*C*m = [wt (CaO) − 0.70 wt (TiO_2_)]/[0.73 wt (Al_2_O_3_) − 0.64 wt (Fe_2_O_3_) + 1.40 wt (Fe_2_O_3_) + 1.87 wt (SiO_2_)](1)
P = [wt (Al_2_O_3_) − 0.64 wt (Fe_2_O_3_)]/wt (SO_3_)(2)
N = [wt (Al_2_O_3_) − 0.64 wt (Fe_2_O_3_)]/wt (SiO_2_)(3)
wt (C_4_AF) = 3.04 wt (Fe_2_O_3_)(4)
wt (C_4_A_3_$) = 1.99 [wt (Al_2_O_3_) − 0.64 wt (Fe_2_O_3_)](5)
wt (C_2_S) = 2.87 wt (SiO_2_)(6)
wt (C$) = 1.70 wt (SO_3_)(7)

According to Equations (1)–(7), BBSC clinkers with different contents of brownmillerite were designed, and were identified as GF-1, GF-2, and GF-3, respectively. Moreover, the mix design of raw materials, theoretical clinker ratio value, and calculated mineral compositions are shown in Table 2. The content of C2S and C$ had few changes, the content of brownmillerite increased and the content of C4A3$ decreased according to the sequence of sample numbers.

#### 2.2.2. Calcination of BBSC Clinker

According to the mix design shown in Table 2, LS, AM, AT, and AH were weighed and evenly mixed. Then, water was added and mixed with powders according to the powder to water weight ratio of 10; wet powders were pressed into a φ120 mm × 10 mm waveform cake. Cakes dried in an oven at 105 ± 5 °C to constant weight were placed on a platinum sheet and calcined in a high-temperature electric furnace at 1330 °C for 30 min. After calcination, the cakes were taken out of quickly and quenched by an air fan to room temperature to obtain the BBSC clinkers with different contents of brownmillerite.

#### 2.2.3. Preparation of the BBSC

A total of 88 wt% of the above prepared BBSC clinkers were mixed with 12 wt% of AH, then ground into powders with a Blaine fineness of 450 ± 10 m^2^/kg by a φ500 × 500 mm test ball mill to obtain BBSC.

BBSC mortars were prepared according to the Chinese standard GB/T 17671-1999: the water to binder ratio (W/B) was fixed at 0.5, and the sand to binder ratio was fixed at 3. Fresh mortars were cast into a 40 mm × 40 mm × 160 mm steel mold, the mold was placed at room condition (20 °C, 50% RH) for 1 d, and then demolded. Demolded samples were cured in a curing box (Cement concrete standard curing box, Huanan Laboratory apparatus Co., Ltd., Wuxi City, China) at 20 ± 1 °C and 95% RH for different ages. BBSC pastes without sand were also prepared to test the hydration products of the BBSC (minerals and micromorphologies of the hydration products), and the W/B was fixed at 0.4 in this case. Fresh pastes were poured into a 20 mm × 20 mm × 20 mm mold, placed at room condition (20 °C, 50% RH) for 1 d, and then demolded. Demolded samples were cured in a curing box at 20 ± 1 °C and 95% RH for different ages. Deionized water was used across the whole experimental process to overcome the influence of impurity ions in the water on BBSC hydration. For these studies, specimens were taken out of the curing box at the designed curing age and cracked by a hammer. The broken pieces of the specimens were kept for a process of stopping hydration by soaking in an excess of ethanol for 24 h and dried for 20 min at 40 °C. Finally, parts of the samples were ground into fine powder with an agate mortar for thermogravimetric analysis (TGA) and XRD.

### 2.3. Test Methods

#### 2.3.1. X-ray Diffraction

X-ray diffraction (XRD) patterns were collected on a Germany Bruker D8 ADVANCE diffractometer with Cu Ka radiation (λ = 1.5406 Å) generated at room temperature at 60 kV and 80 mA. Powders were step scanned from 5° to 70° with a step size and time per step of 0.02° and 0.5 s, respectively. The identification of clinker minerals was implemented by analysis of Bruker EVA V4.2.0.14 search/match software. Rietveld refinement quantitative phase analysis was performed by using Bruker TOPAS 5.0 software through quantitative XRD (QXRD). The refined overall parameters included the emission profile, background, instrument factors, and zero error [21]. The Rwp value of the profile refinement was used to evaluate the quality of the fits in the Rietveld refinement process. Generally, the result is considered as reliable if the Rwp value is less than 10% [21,22]. The ICSD codes of the main minerals in the BBSC clinkers and their hydration products in this study are shown in Table 3 [23,24].

#### 2.3.2. Morphology Analysis

The BBSC clinkers were crushed and molten sulfur was injected into a specific mold to finish casting. The grains of the clinkers were fixed. Then, the sample was ground into a light sheet on the grinding and polishing machine. A total of 1% ammonium chloride solution was used to erode the clinker and the mineral morphology was observed after erosion. The lithofacies of the clinkers were analyzed using a research-grade upright digital material microscope (Axio Scope. A 1) produced by CARL ZEISS in Germany.

The micromorphology of the clinker and hydrated products was analyzed by using the environmental scanning electron microscope (ESEM, Quanta250, FEI, Hillsboro, OR, USA) equipped with energy dispersive spectroscopy (EDS).

#### 2.3.3. Isothermal Calorimetry

The hydration heat release characteristics of BBSC with different contents of brownmillerite were studied. The release of the hydration heat of samples in different plans was measured by a TAM AIR 08279 micro calorimeter produced by TA Instruments from the New Castle, DE, USA. The W/B was fixed at 0.40 according to the Chinese standard GB/T12959-2008 “Test method of hydration heat of cement” and was in accordance with the BBSC pastes.

#### 2.3.4. Thermogravimetric Differential Scanning Calorimeter

TG-DSC was performed by using approximately 50 mg of the resulting powder. Its weight loss was monitored from 50 to 1000 °C with a heating rate of 10 °C/min by a Netzsch STA409 device made by of Germany, while continuously purging with N_2_. The amount of bound water (H), the nature of the hydrates as well as semiquantitative information on crystalline and amorphous hydrate phases were collected by TG analysis. Temperature ranges where specific reactions occurred were obtained from the derivative curve (TG) by differential scanning calorimeter (DSC).

## 3. Results and Discussion

### 3.1. Mineral Composition of BBSC Clinkers

The main chemical compositions of the BBSC clinker are shown in Table 4. At the same calcination condition, the content of Fe_2_O_3_ increased with the increase in the designed C_4_AF content, while the content of SO_3_ decreased with the decrease in the designed C_4_A_3_$.

The clinker lithofacies are shown in Figure 1. The BBSC clinker was mainly composed of leafy and round mineral grains, but there were also some white minerals with different shapes and small gray grains, which could be indexed as C_2_S, brownmillerite, and C_4_A_3_$, respectively [25,26]. Leafy minerals and white minerals increased with the increase in the designed brownmillerite content. Then, a finer morphology of the samples after the analysis of the lithofacies was continued to be observed by using ESEM. Gray round crystal particles in the GF-1 lithofacies corresponded to the well-developed hexagonal C_4_A_3_$ crystals in the ESEM, and there was also a molten liquid phase (brownmillerite). Leaf-like minerals with certain crystal grains in the GF-2 lithofacies corresponded to needle-like and columnar minerals in the ESEM, in which the columnar minerals might be brownmillerite. Coarse needle crystals in sample GF-3 were observed, but hexagonal small crystal particles in C_4_A_3_$ disappeared [4]. The crystal morphology of the main minerals became slender with the increase in the designed C_4_AF, and hexagonal small crystal particles C_4_A_3_$ disappeared, which is consistent with the observation of the lithofacies. It is expected that the increase in iron in raw materials would decrease the eutectic point and increase the liquid phase content, and then increase the solvent mineral brownmillerite during the clinker calcination process, which changed the crystal morphology of the clinker minerals due to the relatively higher calcination temperature for the clinker with higher designed brownmillerite. In addition, the content of C_4_A_3_$ was decreased with the increased content of iron, so hexagonal small crystal particles disappeared with the increase in the designed brownmillerite [26].

The QXRD patterns of the BBSC clinkers with different brownmillerite contents are shown in Figure 2, and the corresponding mineral compositions are listed in Table 5. It can be seen from Figure 2 that the main minerals of the BBSC clinkers were C_2_S, C_4_A_3_$, brownmillerite (C_4_AF, C_2_F, etc.), and a few gehlenites, which corresponded to the mineral design of the BBSC clinker. As shown in Table 5, the main mineral content between the theoretical calculation and QXRD results showed few differences. It showed that the main clinker mineral formation reaction of the whole system was relatively completed at the calcination temperature of 1330 °C. The content of C_2_S ranged from 50% to 54% and the content of brownmillerite gradually increased while the content of C_4_A_3_$ decreased, which also corresponded to the mineral content design of the BBSC clinker.

### 3.2. Hydration of BBSC

A micro calorimeter was used to analyze the hydration heat release of the BBSC clinkers, and the results are shown in Figure 3a. The dissolution of solid particles and the rapid formation of ettringite (AFt) due to the hydration reaction among the AH, C_4_A_3_$, and brownmillerite contributed to the initial higher heat flow [27,28]. After a much shorter induction stage, two significant heat flow humps were clearly visible in the three BBSC systems: the first hump of 9.67 mW/g for GF-1, 6.87 mW/g for GF-2, and 6.75 mW/g for GF-3 at about 5 h; the second hump of 8.83 mW/g at about 7.1 h for GF-1, 5.48 mW/g at about 9.1 h for GF-2, and 4.20 mW/g at about 9.6 h for GF-3. This phenomenon is similar to the reported hydration process of C_4_A_3_$ or brownmillerite in the presence of AH [10,11,12,13,14,27,28], and a two-step hydration reaction was thus proposed [4,29]: the first step at about 5 h, the rapid formation of AFt along with the dissolution of C_4_A_3_$ or brownmillerite and AH; and the second step after 7.5 h, the renewed fast formation of AFt synchronously occurred with a renewed accelerated dissolution of C_4_A_3_$ or brownmillerite, or plenty of Ca(OH)_2_ reacted with AH_3_ and sulfate to produce much more AFt in the Ca(OH)_2_ solution and sufficient gypsum environment. Referred to the hydration mechanism of C_4_A_3_$ or brownmillerite, the hydration reaction of BBSC can be described as Equations (8)–(11),
C_4_A_3_$ + 2C$H_2_ + 34H → AFt + 2AH_3_(8)
C_4_AF + 2C$H_2_ + 30H → AFt + CH + 2xFH_3_ + 2 (1 − x) AH_3_(9)
C_6_AF_2_+ 2C$H_2_ + 35H → AFt +3CH+ 2 (1 + x) FH_3_ + 2 (1 − x) AH_3_(10)
AH_3_ + 3CH + 3C$H_2_ + 20H → AFt(11)

It has been shown that the hydration of brownmillerite would be delayed in the presence of adequate sulfates [10,30]. Therefore, the hydration heat flow decreased by 30% with the increase in the content of brownmillerite in BBSC in the first step. Compared to that of GF-1, the hydration heat flow was further decreased (decreased by 38% and 52%, respectively) and its occurrence time was delayed 2.0 h to 2.5 h with the increase in brownmillerite in BBSC for GF-2 and GF-3 at the second step, respectively. As shown in Table 4, SO_3_ decreased due to the increase in the designed brownmillerite and the decrease in designed C_4_A_3_$ for the BBSC clinker. For GF-1, a relatively higher sulfate level ensured the renewed fast formation of AFt in this case. With the sulfate depletion, monosulfate (AFm) was formed due to the direct hydration of C_4_A_3_$ (Equation (12)) or the AFt transformation,
C_4_A_3_$ + 18H → AFm + 2AH_3_(12)

Meanwhile, brownmillerite hydration was also initiated with the sulfate depletion in this step [7,19]. Interestingly, another obvious hydration flow hump at about 120 h was observed for sample GF-3 in Figure 3b, which may be the C_2_AS·8H_2_O formation [20,31,32], as shown in Equation (13),
C_2_S + AH_3_ + 5H → C_2_AS·8H_2_O(13)

The total heat release of BBSC hydration is shown in Figure 3b. The total heat release at early ages (7 d) for the three BBSC systems was about 200 J/g, which was lower than that of sulfoaluminate cement and almost equal to that of low-heat cement [1,4]. It was noted that the total heat release of GF-2 (below 250 J/g) was lower than that of GF-1 and GF-3 after 120 h. The total heat release of BBSC before 120 h was lowered with the increase in brownmillerite content, which would further increase after 120 h. A much higher total heat release for GF-3 was observed, which was in accordance with the heat flow result shown in Figure 3a, and the formation of C_2_AS·8H_2_O might contribute to the higher later heat release.

The hydration products of the BBSC paste cured for different times were tested by XRD. As shown in Figure 4, the main hydration products of BBSC consisted of AFt, C_2_AS·8H_2_O, unhydrated C_2_S, and few low-sulfur AFt (AFm) [4,18,33,34,35]. Furthermore, AFm, AH_3_, and FH_3_ were also present, however, they were not obviously observed in the XRD patterns due to their gel state and poor crystal development. It can be found that the peak intensity of AFt located at 25° (2θ) for each sample after curing for 7 d was the strongest, and it decreased with the increased amount of brownmillerite. The diffraction peak located at 8° (2θ) could be identified as C_2_AS·8H_2_O, which could be observed in GF-1 after 21 d of curing, in GF-2 after 7 d of curing, and in GF-3 after 3 d of curing, respectively. That is, the higher the brownmillerite content in the BBSC, the earlier the C_2_AS·8H_2_O formed. The diffraction peak intensity of C_2_AS·8H_2_O gradually became stronger with the prolonged curing age, and it also rose with the increased content of brownmillerite in BBSC. These results correspond to the above hydration heat release analysis. Meanwhile, the peak of unhydrated dicalcium silicate gradually weakened with the prolonged curing age, indicating the hydration of C_2_S and the hydration reaction shown in Equation (13).

The ESEM images of BBSC after curing for different times are shown in Figure 5. After 1 d of hydration, relatively loose structures were observed, and the main hydration products were needle-like AFt, flocculent products, and flaky AFm [10]. The BBSC pastes gradually became denser with the prolonged hydration time. After curing for 3 d, long-needled AFt could be observed in both GF-1 and GF-2. GF-1 had more long-needled AFt, while some short-needled AFt and flaky AFm still existed in GF-2. In GF-3, hexagonal and flaky AFm could be seen. The results showed that the shape of AFt was obviously affected by the content of brownmillerite in BBSC. After curing for 21 days, the GF-1 paste became denser, with many small grains and antler-shaped hydration products originating from the small grains (see in the yellow box of Figure 5j). The structure of the GF-2 paste was more loose than that of GF-1, but many antler-shaped hydration products with relatively sharp ends also appeared. With regard to GF-3, the paste structure was more loose than that of the other two samples. A large amount of interwoven long string-like hydration products appeared (see in the yellow box of Figure 5k). With the change in the brownmillerite content, hydration products with different morphologies appeared. After 28 days of curing, many comparatively thick grass-shaped hydration products with sharp ends were produced in GF-1. Many small grains also appeared in GF-2 and grass-shaped hydration products generated by small grains became relatively longer and thinner. In GF-3, many grass-shaped hydration products came into being. The ESEM results showed that the morphology of the hydration products at different curing times changed obviously with the increase in brownmillerite in BBSC, especially in the later curing stage. The grass-shaped hydration products developed obviously and grew luxuriantly with the increase in brownmillerite content in BBSC.

EDS was used to scan the elements in the selected area of the hydrated sample, where the results are shown in Figure 6 and data from the EDS results are listed in Table 6. Elements and their content for hydration products with different morphologies after 1 d and 28 d of curing had no obvious difference, which could all be identified as AFt. That is, the increase in brownmillerite might change the morphologies of AFt as grass-like hydration products were also the AFt enriched in iron [4].

The hydration products of the samples cured for 1 d, 3 d, and 7 d were analyzed by TG-DSC. All samples had similar weight loss results, thus the TG-DSC pattern of GF-2 was selected as the representative result and is shown in Figure 7. A two-stage weight loss process was observed: the obvious weight loss with the endothermic peak between 100 °C and 200 °C corresponded to the decomposition of AFt [36]; another obvious weight loss with an endothermic peak between 400 °C and 500 °C corresponded to the decomposition of CH. Detailed weight losses at each stage for all samples are listed in Table 7. Between 100 °C and 200 °C, the weight loss of GF-2 after curing for 1 d and 3 d was higher than that of the other samples, indicating the formation of a higher content of AFt, which would be conducive to the mechanical performance development. At 7 d, the weight loss of GF-1 and GF-2 decreased due to the rapid consumption of gypsum in the early reaction and AFt crystals were transformed into AFm in the condition of insufficient gypsum. However, the weight loss of sample GF-3 increased with the prolonged curing time, indicating that the increase in the iron content can stabilize the formation and crystal development of AFt, thus, it might be conducive to the later strength development. Between 400 °C and 500 °C, the weight loss increased with the increase in brownmillerite at 1 d due to the increased content of CH, according to Equations (9) and (10). After curing for 1 d, the weight loss of GF-2 was lower than that of the other samples, indicating that the reaction of Equation (11) occurred at this stage. The CH produced by GF-2 hydration and the remaining gypsum made the reaction of Equation (11) reach the best state. Relatively, insufficient CH in GF-1 and excessive CH in GF-3 might occur.

In short, the hydration process of BBSCs can be clearly divided into two steps: the first one corresponds to the hydration of C_4_A_3_$ and brownmillerite at an early age, and the second one involves the contributions of belite and brownmillerite hydration [36]. Brownmillerite might promote the formation of C_2_AS·8H_2_O according to Equations (9), (10) and (13) and influence the morphologies of AFt.

### 3.3. Mechanical Performance of BBSC

Mechanical performances of BBSC mortars after curing for 1 d, 3 d, 7 d, and 28 d were tested and the results are shown in Figure 8. The flexural strength and compressive strength of the BBSC mortars were relatively higher after 1 d and 28 d. The decreased flexural strength of the BBSC mortars after curing for 3 d compared to that of curing for 1 d was observed. The flexural strength of GF-2 and GF-3 cured for 7 d started to increase compared to that of curing for 3 d, while the flexural strength of GF-1 continued to decrease compared to that of curing for 3 d. After 28 d of curing, the flexural strength of GF-2 and GF-3 was slightly higher than that of curing for 1 d, while the flexural strength of GF-3 increased more than that of curing for 1 d. As shown in Figure 8b, the compressive strength of GF-1 mortars had little increase with the prolonged curing times at the early stage (before 7 days); in contrast, the compressive strength of GF-2 and GF-3 mortars slowly increased with the prolonged curing times. It has been reported that the expansion performance of AFt was dramatically influenced by the hydration environment [37]. AFt and AH_3_ were rapidly formed due to the fast hydration of C_4_A_3_$ or brownmillerite and AH according to Equations (8)–(10) at the early hydration stage of BBSC, which provided the high early strength of the hardened BBSC pastes. At this time, CH, AH_3_, and FH_3_ were formed. Then, according to Equation (11), plenty of CH reacted with AH_3_ and sulfate to produce significantly more AFt in the environment of the CH solution and sufficient gypsum, which generated a lot of expansion energy and might have decreased the mechanical performances of the hardened BBSC. As the hydration reaction proceeded, C_2_S participated in the hydration reaction according to Equation (13). As shown in Figure 5, the interwoven grass-like AFt increased with the increased content of brownmillerite, which filled the pores of the hardened pastes and increased the toughness of the BBSC mortar, thus improving the flexural strength [36,37,38]. As mentioned in Figure 4, the higher the brownmillerite content in BBSC, the earlier the C_2_AS·8H_2_O formed [31,32], which is why the compressive strength of the GF-2 and GF-3 samples increased faster than that of sample GF-1. It could also be observed that the 28 d compressive strength of the BBSC mortar had slightly decreased with the increase in brownmillerite content, which might be because the many grass-like products decreased the compactness of the structure. In general, the development law of mechanical strength is consistent with the hydration reaction process, and the increase in the brownmillerite content contributes to the stability improvement in the flexural strength and the stable growth in the compressive strength of BBSC.

## 4. Conclusions

The main findings of this study can be summarized as follows:

1. BBSC clinkers with different contents of brownmillerite (15~25%) were successfully prepared through calcination at 1330 °C for 30 min. The increase in the designed brownmillerite content would decrease the eutectic point and increase the liquid phase content, which changed the crystal morphology of the clinker minerals.

2. The increase in brownmillerite in BBSC decreased the hydration exothermic rate and delayed its occurrence time. Meanwhile, the increase in brownmillerite in BBSC might promote the C_2_AS·8H_2_O formed, leading to the second increase in the hydration heat release.

3. The hydration products of BBSC mainly consisted of AFt, C_2_AS·8H_2_O, AFm, CH, AH_3_, and FH_3_. C_2_AS·8H_2_O increased with the prolonged curing times and the increase in brownmillerite. AFt enriched in iron with interwoven grass-shaped morphologies were observed; these formed earlier and developed more with the increase in brownmillerite.

4. The formation of AFt enriched in iron improved the flexural strength shrinkage, but had little effect on the compressive strength. The increase rate in the compressive strength of BBSC will increase with the increased brownmillerite content in BBSC.

## Figures and Tables

**Figure 1 materials-15-04344-f001:**
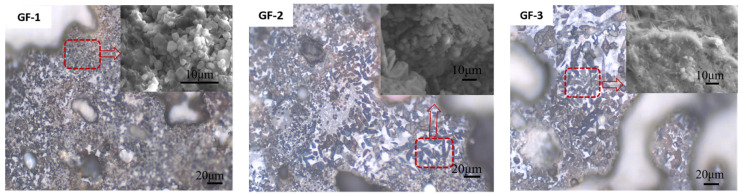
The morphology of the BBSC clinkers (inserts are the ESEM patterns).

**Figure 2 materials-15-04344-f002:**
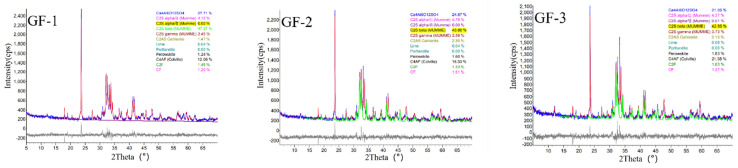
The QXRD patterns of the BBSC clinkers.

**Figure 3 materials-15-04344-f003:**
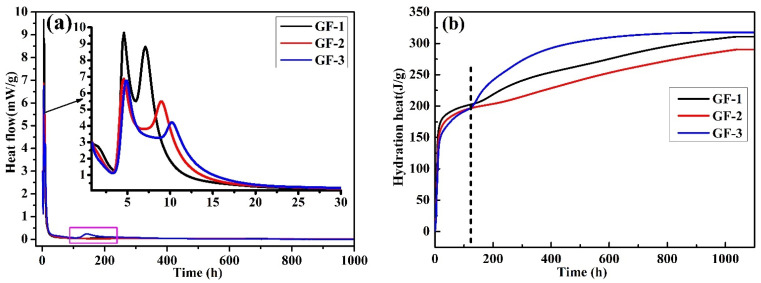
(**a**) The hydration heat flow and (**b**) total hydration heat of the BBSC.

**Figure 4 materials-15-04344-f004:**
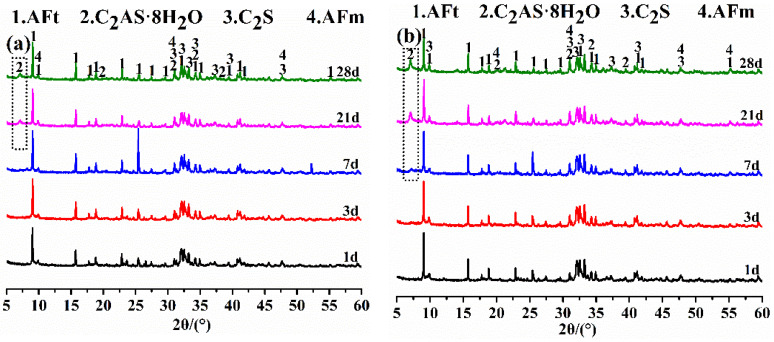
The XRD patterns of the BBSC hydration products: (**a**) GF-1, (**b**) GF-2, and (**c**) GF-3.

**Figure 5 materials-15-04344-f005:**
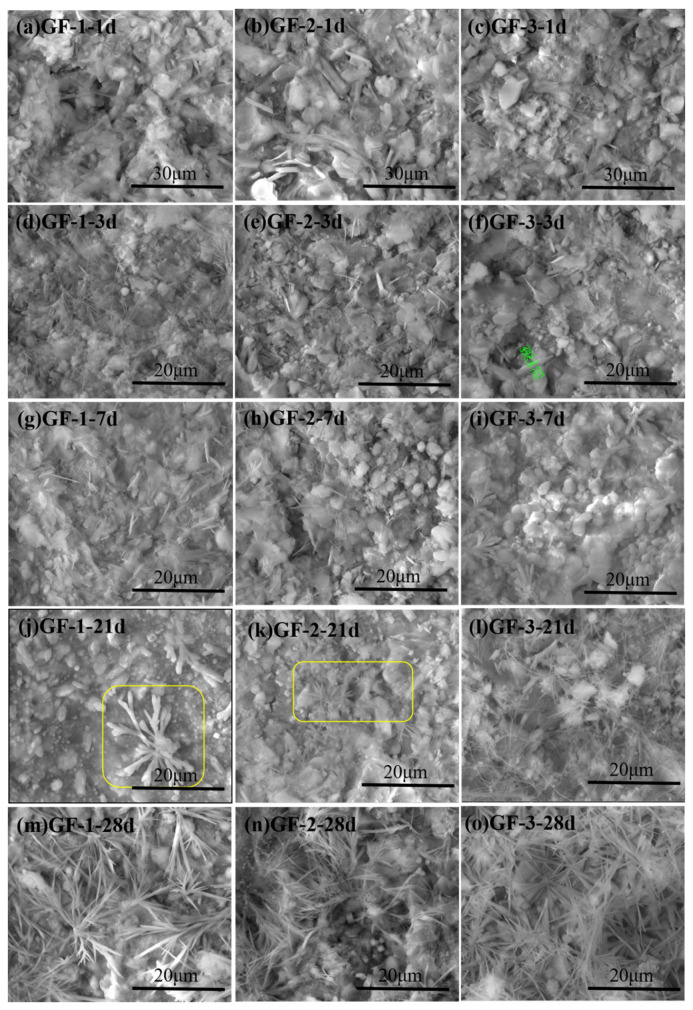
The ESEM images of the BBSC pastes after curing for (**a**–**c**) 1 d, (**d**–**f**) 3 d, (**g**–**i**) 7 d, (**j**–**l**) 21 d, and (**m**–**o**) 28 d.

**Figure 6 materials-15-04344-f006:**
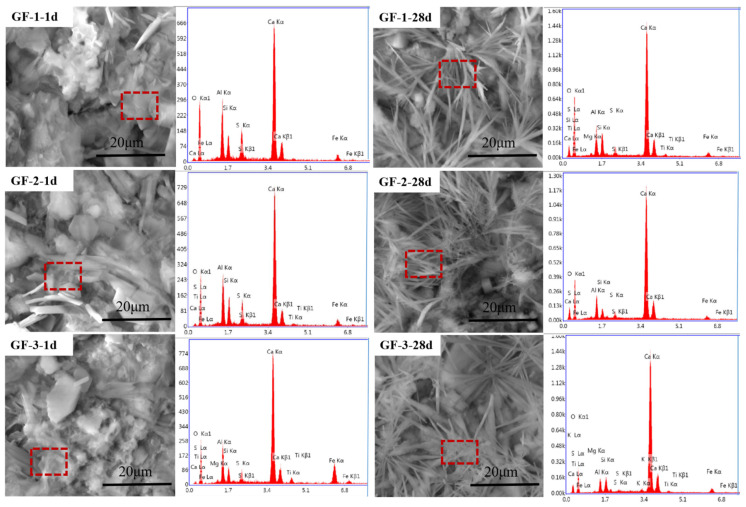
The ESEM-EDS images of the BBSC paste.

**Figure 7 materials-15-04344-f007:**
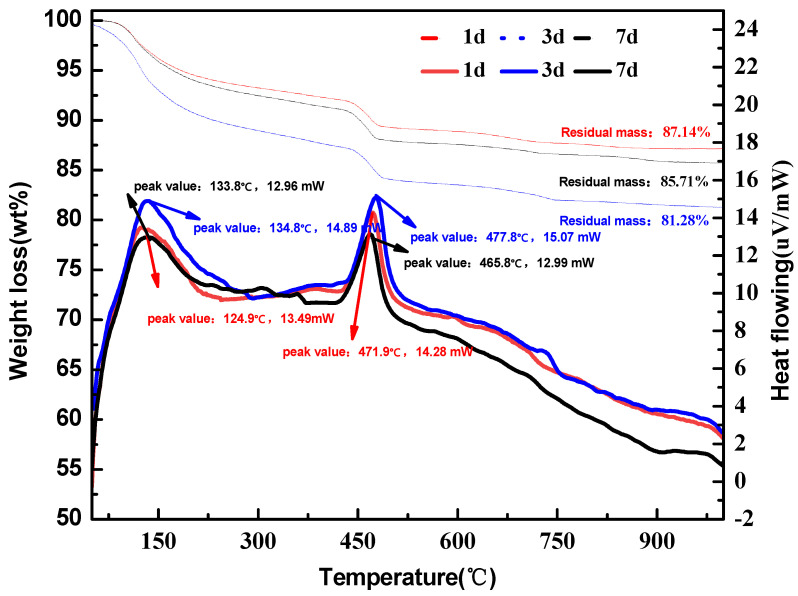
The TG-DSC of sample GF-2.

**Figure 8 materials-15-04344-f008:**
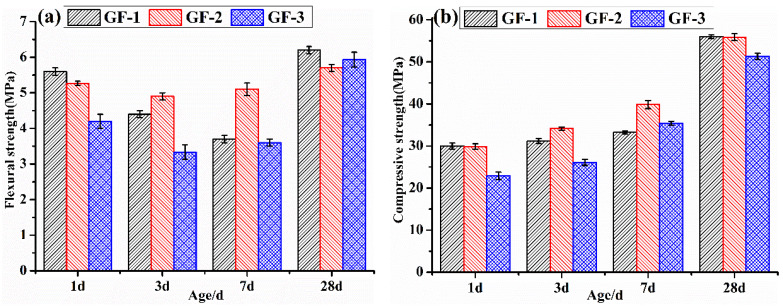
(**a**) The flexural and (**b**) compressive strength of the BBSC mortars.

**Table 1 materials-15-04344-t001:** The main chemical compositions of raw materials (wt%).

Materials	Loss	SiO_2_	Fe_2_O_3_	Al_2_O_3_	CaO	MgO	TiO_2_	SO_3_	∑
LS	43.37	1.03	0.16	0.26	54.13	0.44	-	-	99.39
AT	14.06	37.20	1.88	43.15	0.57	0.41	-	0.05	97.32
AM	9.70	39.18	24.46	19.74	1.69	0.60	3.87	-	99.24
AH	8.47	4.05	1.02	0.41	33.85	5.12	-	46.60	99.52

**Table 2 materials-15-04344-t002:** The mix design of the raw materials, theoretical clinker ratio value, and calculated mineral composition of the BBSC clinkers.

Sample	Raw Materials (wt%)				Clinker Ratio Value			Main Minerals Composition (wt%)			
	LS	AM	AH	AT	Cm	P	N	C_4_A_3_$	C_2_S	C_4_AF	C$
GF-1	60.60	12.30	7.10	20.00	0.98	3.01	0.81	28.76	51.38	15.08	1.82
GF-2	61.20	17.60	6.20	15.00	0.98	2.89	0.69	24.21	51.76	20.04	1.80
GF-3	62.00	22.90	5.10	10.00	0.97	2.85	0.57	19.66	50.16	25.04	1.56

**Table 3 materials-15-04344-t003:** The ICSD of the main minerals in the BBSC clinkers and their hydration products in this study.

Mineralogical Phase	ICSD
α-C_2_S	81097
β-C_2_S	81096
γ-C_2_S	81095
C_4_A_3_$	9560
C_4_AF	9197
Perowskite	62149
C_2_AS	87144
Anhydrite	16382
Brownmillerite	80869
Portlandite	15471
Ettringite	16045
AFm	/
C_2_AS·8H_2_O	69413

**Table 4 materials-15-04344-t004:** The main chemical compositions of the BBSC clinkers (wt%).

Sample	Loss	SiO_2_	Al_2_O_3_	Fe_2_O_3_	CaO	TiO_2_	SO_3_	MgO	∑
GF-1	0.26	17.85	16.67	6.17	50.89	1.55	4.43	1.81	99.63
GF-2	0.36	17.50	16.06	7.09	51.04	1.79	3.81	1.51	99.16
GF-3	0.46	17.34	14.89	8.97	51.18	1.83	3.17	1.61	99.45

**Table 5 materials-15-04344-t005:** The main mineral compositions of the BBSC clinkers from the QXRD results (wt%).

Sample	Theoretical Value				Calculated Value by QXRD		
	C_4_A_3_$	C_2_S	C_4_AF	C$	C_4_A_3_$	C_2_S	Brownmillerite
GF-1	28.76	51.38	15.08	1.82	27.71	54.82	14.71
GF-2	24.21	51.76	20.04	1.80	24.87	52.04	19.14
GF-3	19.66	50.16	25.04	1.56	21.05	50.55	24.28

**Table 6 materials-15-04344-t006:** The elemental structure of the cement paste.

Sample	1 d					28 d				
	Al_2_O_3_	SiO_2_	SO_3_	CaO	Fe_2_O_3_	Al_2_O_3_	SiO_2_	SO_3_	CaO	Fe_2_O_3_
GF-1	1.54	0.91	1.21	0.85	2.42	1.59	0.93	1.25	0.87	2.48
GF-2	1.53	0.90	1.21	0.84	2.43	1.63	0.96	1.28	0.90	2.55
GF-3	1.65	0.97	1.30	0.91	2.51	1.67	0.98	1.31	0.92	2.61

**Table 7 materials-15-04344-t007:** The weight loss of the sample derived from the TG-DSC results (wt%).

Sample	100–200 °C	400–500 °C	Residual Mass
GF-1-1d	−5.67	−4.47	87.58
GF-1-3d	−9.92	−6.85	80.06
GF-1-7d	−7.36	−5.62	84.69
GF-2-1d	−6.08	−4.84	87.14
GF-2-3d	−10.67	−5.52	81.28
GF-2-7d	−7.28	−4.91	85.71
GF-3-1d	−4.34	−6.90	88.57
GF-3-3d	−6.36	−12.20	80.26
GF-3-7d	−8.42	−10.83	79.04

## Data Availability

Data will be made available upon reasonable request.

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
