# Peer review of "Preparation and Hydration of Brownmillerite-Belite-Sulfoaluminate Cement"

_materials, 2022, doi:10.3390/ma15124344_

Round 1
Reviewer 1 Report
Dear author
I have completed the review of your manuscript, after the terms mentioned were correctedand, so i recommend the paper for publication. I have just one concern about formatting, but i suppose the publishing section of Materials will take care of that.
Best Regards
Author Response
Dear editor ,
Thank you very much for your careful review of the manuscript. Furthermore, we have modified the writing, and the details were outlined in red color in the revised manuscript.

Reviewer 2 Report
Dear author,
the result from this research was good to be published. however, the arrangement of the discussion was not well organized. the SEM image should be clearly label and the area of the discussion should be clearly highlighted. the depth of the discussion still can be improved.

Author Response
Dear editor,
Thanks for your efficient work. We are thankful to your critically examining and suggestion for the work. The suggestions have helped us in improving the manuscript quality. The whole manuscript has been modified. The point-wise response to your comments is presented as follows.Please see the attachment.
Best Regards,
Xuemei CHEN

Reviewer 3 Report
Comments to the authors:
1. Mineral compositions of BBSC clinkers are suitable to be calculated, based on the Bogue equations, using a computerized optimization software in which to impose clear compositional criteria and from the point of view of the values ​​of the properties of interest;
2. Only the three samples GF1, GF2, GF3 are partially sufficient to draw clear conclusions from the point of view of the composition-structure-properties correlations;
3. The chemical-structural and thermal analyzes involve a very large volume of information, difficult to follow and correlate, therefore it is preferable to be divided eventually into 2 papers for a larger number of samples;
4. There are certain contradictions in interpretation such as … lines 355-358 with the conclusions from lines 377-379.
Author Response

(The authors gave the same response as above.)

Round 2
Reviewer 2 Report
Figure 1 : Scale bar for GF-1, 10um was not equal with GF2 and GF3
Error $ was still found in the text eg Line 40, 82,87, 219, 233 and many more
Font size Line 84-89 is not the same with others.
Author Response
Dear reviewer,
Thank you very much for your efficient work and valuable advice,which indeed helped us in improving the manuscript quality. The whole manuscript has been modified. The point-wise response to your comments is presented as follows.Please kindly review the attachment, thanks.
Best regards,
XM Chen

This manuscript is a resubmission of an earlier submission. The following is a list of the peer review reports and author responses from that submission.
Round 1
Reviewer 1 Report
The main aim of this paper is to prepare different brownmillerite-belite- sulphoaluminate cements and to study the hydration mechanism of those systems. Authors use mainly x-ray diffraction and Rietveld method to perform their characterization. Moreover, to study the hydration process they use the x-ray diffraction (but only in a qualitative way), isothermal calorimetry, SEM analysis and mechanical and flexural strengths. However, the studies are not consistent as they use different w/b ratios for their studies and the x-ray diffraction characterization is not complete, Rietveld analysis are required. Moreover, the SEM analysis is not conclusive as they use only 5 points of analysis per sample (in the same region) to obtain the quantitative values.
On the other hand, there are some protocols and explanations that are not clear and they have to revise them. I have added all my comments in the pdf file (that I attach) that should be addressed but I considered that more experimental work (for instance mayor revision) has to be included in the article before being published.

Author Response
We are thankful to the reviewer for critically examining and appreciating our work. The suggestions made by the reviewer have helped us in improving the manuscript quality.
As suggested by the reviewer, the whole manuscript has been modified.
The point-wise response to the reviewer comments is presented in the following.
Please see the attachment.

Reviewer 2 Report
The experimental part of the study looks interesting. However, the discussion of the results obtained must be really improved before the paper deserves publication. Authors do not compare the results obtained with previous ones. In this sense, even if the BBSC are not already evaluated in depth, authors must compare the results obtained with others already published in order to better explain the innovation of their results. For example, there is not any reference mentioned in the discussion of the mechanical results obtained and this is not acceptable in a research manuscript. Moreover, English style must be improved, mainly in the first part of the manuscript.
Other specific comments:
- The number of references considered in the study seems to be quite scarce.
- Lines 60-61: Bogue equations were mainly developed for ordinary Portland cements (CEM-I type). Why did you use these equations in your special clinker?
- Figure 3: the increase around 5 days of hydration related to C2AS·8H2O formation is a little bit surprising. Authors mention reference 18 to explain this phenomenon but in this reference there is not any calorimetry result.
Author Response
The authors are thankful to the reviewer for appreciating our work.
The suggestions by the reviewer have immensely helped us enhance the quality of the manuscript.
The point-wise response to the reviewer comments is presented in the following.
Please see the attachment.

Reviewer 3 Report
The manuscript describes an interesting study of the Hydration of Brownmillerite-Belite-2 Sulphoaluminate Cement. Overall the manuscript is well-written.
Suggestions for Authors
1: The main novelty is Hydration of Brownmillerite-Belite-2 Sulphoaluminate Cement, but this technique is not clearly presented and it is not made clear what novelty this brings.
2: All tables are missing error measure, and as they are all close values its information is relevant.
3- The figures 1, 2, 4, 5 and 6 are very small and must be improved in size and quality (it´s not possible to read them).
4- In figures 2 and 4 the authors identify different phases but it is not clear in the spectra. The figures need to be analyzed in depth, clearly identifying all hydrated phases and providing a scientific explanation of the differences found in both and why.
5- Of Table 3, how the fluctuation of SiO2 influences the final result of the Hydration of Brownmillerite-Belite-2 Sulphoaluminate Cement.
6- The figure 7 needs to have its color palette changed, for better analysis
7- Please check English.
Author Response
The authors are thankful to the reviewer for appreciating our work.
The suggestions by the reviewer have immensely helped us enhance the quality of the manuscript.
The answer to the comments has been given below.
Please see the attachment.

Round 2
Reviewer 1 Report
I cannot accept the publication as they did not carry out all my proposed experiments.
For instance:
-My comment about SEM STUDY: This study has not any sense. Because there are many phases in the hydrated pastes and you are only using 5 points that are in the same area to do the average. More points taken from different regions are needed if you want to show these results.
Additionaly, the SEM analysis are semiquantitative.
Moreover, you do not need to show all the points/numbers with the average (of a reasonable number of points) is enough
They said that…”They did more research” but they did not show that.
-My comment about QXRD: QXRD is also neccesary to obtain the phase contents.
They said that ”It is difficult” but they did not try
Moreover, they did not answer all my questions. In some cases, they did delete some sentences of the original text in case not give some explanations about this. I consider this very bad praxis.
-They said in the original version: Hardened BBSC pastes curing for different ages were crushed, collected and stored in anhydrous ethanol to stop hydration, then dried at 80 ℃ for 24 h before XRD test.
My question was: this is a very high temperature; how do you know that you are not altering the main hydrated phases? For instance, Ettringite. Please give details or references to explain why you use this temperature
Now, they have delete this sentence, but this is very important point as 80ºC is a very high temperature and the samples can decompose.
Reviewer 2 Report
Authors have made several modifications improving the quality of the manuscript.